# Identifying Key Selection Criteria for Smart Building Technologies in the United Arab Emirates Prisons

**Mohammed Abdulla Mohammed Mesfer Aldhaheri** *, **Bo Xia** and **Madhav Nepal**

School of Architecture and Built Environment, Faculty of Engineering, Queensland University of Technology, Brisbane, QLD 4000, Australia
* Correspondence: m.aldhaheri@hdr.qut.edu.au

**Abstract:** The selection of an appropriate smart building technology has been a challenge for stakeholders, because no specific selection criteria are currently available. This study aimed to identify the potential selection criteria for the selection of smart building technologies for prison buildings in the United Arab Emirates. A questionnaire survey was conducted to evaluate the relative importance of smart building technologies and the specific selection criteria. 238 experts from the public and the private sector with rich experience in the construction and prison industry participated in the survey. The data obtained were analyzed for descriptive statistics and the Mann-Whitney U test was conducted to compare the responses of the government and private sector respondents. Cronbach's coefficient was estimated using reliability analysis. Finally, exploratory factor analysis was performed by Principal Axis Factoring (PAF) to extract the contributing factors and was further improved by varimax rotation using SPSS. To evaluate the appropriateness of the factor extraction, the Kaiser-Mayer-Olkin (KMO) measure of sampling accuracy and Barlett's test of sphericity were conducted. The results demonstrated that most participants thought that the safety and security, anti-hacking capability, high working efficiency, and durability of the new smart building technology were very important. 14 listed selection criteria were extracted into three factors by factor analysis explaining 50.585% total variation. Regarding smart building technologies, fire protection was mostly voted by the participants followed by video surveillance and heat, ventilation, and air-conditioning system (HVAC). This study is a novel research study identifying the key selection criteria for the selection of important smart building technologies and would be helpful for a broad audience.

**Keywords:** smart building technology; selection criteria; HVAC; principal component analysis; UAE; prisons

## 1. Introduction

Buildings used to be a crucial component of complex ecosystems as they share some unique characteristics with living organisms such as energy cycling, information sharing, and interaction with the outer environment. While compared with traditional buildings, intelligent or smart buildings experience better interaction with their residents and the environment. The smart building concept was pioneered during the 1980s in the USA and evolved quickly throughout the world [1]. A smart building has a unique set of integrated systems that perform different functions of the building such as energy and power management, building security, and surveillance very smoothly thus keeping energy losses to a minimum to maintain sustainability [2]. A truly smart building not only provides safe, energy-saving, environmentally friendly, and convenient services but also incorporates situational awareness, which allows it to proactively respond to the presence of people and adapt to changing circumstances throughout the day using actuators and devices that control engineering systems such as heating, lighting, and air conditioning [3]. Smart buildings have been widely adopted in various parts of the world. Some popular smart buildings include the Mirage in Las Vegas, Three Logan Square in Philadelphia, the Los

Angeles Convention Center, and the City of San Francisco Public Utility Commission headquarters [4].

To make buildings smart, various smart building technologies have been used globally. The most commonly known smart building system is a building management system (BMS), which manages buildings and their interaction with users. The BMS facilitates proactive management of various mechanical, electrical, and plumbing systems, and monitors and modulates the performance metrics of a building [5,6]. The BMS is further enhanced by integrating modern technologies such as the internet of things (IoT) and wireless sensor technology [7]. Besides BMS, building automation systems (BAS) integrate and automate various functions of a building whereas energy management systems (EMS) are used to monitor the power transmission system and have been widely researched due to their ability to integrate and manage several kinds of automation technologies within the building [8]. Smart buildings also require heating, ventilation, and air conditioning (HVAC) systems, lighting control systems, access control systems, video surveillance, and facility management systems [9].

Selecting an appropriate smart technology for a specific building has always been an interesting subject for researchers and stakeholders. The most prominent of these criteria include reliability, security, working efficiency, integrity, and market potential [10]. Another critical selection criterion is cost-effectiveness as smart buildings are considered to be associated with lower maintenance and operational costs because of automation. It is estimated that while smart buildings may have a 25% higher initial cost compared with conventional buildings, they could generate an average of 38% operational and maintenance efficiency over the next 30 years. Some other criteria for selecting a smart building technology may include durability, and credible suppliers [11]. Intelligent fresh air supply along with thermal control is also deemed critical when selecting smart building technologies [12]. Another crucial criterion is the environmental sustainability of the smart buildings which is considered a key criterion aimed to enhance working efficiency, and reduce energy losses while maintaining residents' living standards and improving environmental stability [2]. Omar [13] stated that a diversity of selection criteria might be considered for selecting and evaluating smart buildings such as safety, working efficiency, economics, and living comfort.

It should be kept in mind that the selection criteria for the smart building technologies depend on the type of the smart building whether it is a commercial, residential or correctional facility. Smart prisons are simply smart buildings designed for prison facilities to monitor and control inmates' behavior. Knight and Van De Steene [14] define a smart prison as a structure that has an automatically controlled perimeter wall and lighting, optimum water and energy control, remote monitoring, automated data collection on inmates and the building, use of RFID to track and control violent inmates, and interception of illegal calls. Various smart building technologies and prison concepts are being developed to make smart prisons more efficient. For instance, the Belgium Prison Cloud is a smart digital system that is used to monitor inmate activities after release and helps in rehabilitation into society [15]. In their study, Cynthia, et al. [16] demonstrated the usefulness of IoT technology in developing prisoner escape alert and prevention systems. Hence, a smart prison building needs suitable selection criteria to choose smart building technologies to enhance its unique functions such as monitoring and control of inmates.

Unfortunately, a set of selection criteria for smart building technologies for smart prison buildings is missing from the current body of knowledge. Most of the current critical criteria such as energy efficiency, affordable maintenance support, and maintenance and operational costs are mainly designated for residential and commercial smart buildings. Therefore, this study aimed to identify the key selection criteria for suitable smart building technologies that could be implied in prison buildings. In particular, as smart building technologies have been rarely implemented in UAE prisons and there is no specific standard for the selection of smart building technology, this study was designed to assist UAE prison administrators in selecting appropriate smart building technologies for prison buildings.

For this purpose, a broad questionnaire survey was conducted in UAE to collect experts' opinions on (1) the most critical selection criteria of smart building technologies that could be implemented in the UAE prisons, and (2) the importance of various smart building technologies which are currently used in the UAE prisons. Exploratory factor analysis was then performed to identify groups of those selection criteria.

## 2. Materials and Methods

To address the research gap, semi-structured interviews were first conducted with experts from the prison and construction sectors for their opinions on smart building technologies that were currently being used in UAE prisons and the potential criteria for their selection. After that, a comprehensive online questionnaire survey was conducted to obtain the opinions of experts from the prison industry about (1) the key selection criteria that should be considered while selecting an appropriate smart building technology for prison facilities in the context of UAE prisons, and (2) the importance of various smart building technologies for prison buildings in UAE. After the questionnaire survey, explorative factor analysis was performed to group the selection criteria into different groups based on the Eigenvalues.

### 2.1. Questionnaire Design and Data Collection

The questionnaire was designed based on (1) the Building Intelligence Framework (BIF), which highlights distinguishing factors to separate smart buildings from traditional buildings and (2), the Intelligent Building Survey (IBS) which provides the reasoning of BIF with the empirical evidence; both have been extensively used in the study of smart and intelligent buildings [17]. The Building Intelligence Framework (BIF) is now one of the most popular models for analyzing smart building technologies and has been validated under various environments and conditions [18–20]. In addition, the Intelligent Buildings Criteria Selection (IBCS) model was used in designing the survey instrument for the identification of decision criteria. The model has not been tested or validated but remains an ideal framework for assessing the criteria for smart buildings [13].

Initially, semi-structured interviews were conducted involving 14 experts from prison and construction sectors for their opinions on smart building technologies that were currently being used in UAE prisons and the potential criteria for their selection (unpublished data). After conducting a thematic analysis of the interviewees' responses and literature review, 15 smart building technologies and 14 criteria on how to select these technologies were listed in the questionnaire. The questionnaire contains three sections. The first section included questions on the respondents' personal and socio-economic information such as age, gender, employment status, education level, type of organization, and the number of years with the organization. The next section was designed to rank the smart building technologies being used in prison facilities in UAE based on the Likert scale ranging from 1 to 5 (where 1 represented not important at all, and 5 represented extremely important) regarding their importance. The last section explored the opinion of respondents on the relative importance of different criteria for the selection of smart building technologies in the UAE prisons based. Responses to questions in each section were measured on a Likert scale ranging from 1 to 5 (where 1 represented not important at all, and 5 represented extremely important) [21,22].

To validate the survey instrument, a pilot study was conducted with five respondents to identify areas that the respondents find difficult to apprehend. Changes were made where necessary to ensure that the survey instrument is clear and suitable for its purpose.

Potential participants were identified mainly through personal resources and databases from Abu Dhabi Police Head Quarters. Also, LinkedIn profiles of contractors, engineers, consultants, and government officials were skimmed and scrutinized for participation in the survey. The selective respondents with good knowledge of the smart building techniques having a minimum of three years of experience in dealing with smart building technologies were approached to minimize biases. Invitation letters briefly informing the

participant about the goals and scope of the study were sent out to 400 target respondents to officially invite them to participate in the survey through the Qualtrics online platform. Follow-up reminders were sent after every two weeks to those who failed to respond in time to make sure all participants responded to all the questions completely. Finally, 238 responses were collected online within a 40 day timeframe, demonstrating a 60% response rate, which is encouraging as supported by many studies [23,24]. The respondents included contractors, project managers, engineers, government officials, and members of the public. This diversity of respondents enhanced the richness of the data collected.

### 2.2. Data Analysis

In this research, data collected from the survey was first analyzed for normality check by the Shapiro-Wilk test [25]. The data were further subjected to descriptive statistics, and Mann-Whitney U was performed to separate the responses of government and private respondents using the Statistical Package for the Social Sciences [26]. The nonparametric effect size was calculated by the following formula:

$$r = Z/\sqrt{n} \tag{1}$$

where $r$ = effect size coefficient, $n$ = number of respondents in government and private groups.

The value of $r$ ranges from $-1$ to $1$ and Cohan classified it into three categories each for positive and negative coefficients [27] i.e., 0.1–0.3 = small effect, 0.3–0.5 = moderate effect, and 0.5–1.0 = large effect whereas for negative effect size coefficient, $-0.1$ to $-0.3$ = small effect, $-0.3$ to $-0.5$ = moderate effect, and $-0.5$ to $-1.0$ = large effect. Since most of the Z-scores were negative, therefore the absolute value of the effect size coefficient was considered.

Cronbach's alpha was estimated by reliability analysis to evaluate the authenticity of the respondents' opinions. Exploratory Factor Analysis (EFA) was performed on the data obtained from each variable to extract the latent factors which contributed significant contributions to variation among the tested variables. To evaluate the appropriateness of the factor extraction, the Kaiser-Mayer-Olkin (KMO) measure of sampling accuracy and Barlett's test of sphericity were conducted. The KMO ensures that the observed correlation coefficients and partial correlation coefficients are very less whereas Bartlett's Test of sphericity is used to check if there is a certain redundancy between the variables and that at least two variables are correlated [28,29]. KMO measure test showed the value of 0.920 for the selection criteria of the smart building technologies which is greater than the acceptable limit of 0.60 whereas Barlett's test of sphericity confirmed that the data was suitable for factor extraction ($\chi^2$ = 1453.381, df = 91, $p < 0.01$) [30]. Principal Axis Factoring (PAF) was used in this study to extract the factors which is a widely used method for heteroscedastic data [31]. Following factors extraction by PAF, the varimax method of rotation was applied to more accurately group different selection criteria.

### 3. Results

#### 3.1. Sociodemographic Information of the Participants

The socio-demographics of the participants are illustrated in Table 1. Out of 238, 84.9% of participants were males compared to only 15.1% of females. Concerning age, the higher proportion of participants was the aged between 31 and 40 years old (50.4%) followed by respondents aged between 41 and 50 years (22.7%). Only 0.8% of participants of age above 60 years participated in the survey. The survey participants belonged to diverse ethnic groups but most of them were UAE nationals (87.0%). People from other countries such as India, the UK, the USA, France, and Somalia working in the UAE also participated in the survey (13.0%). Most of the participants were based in Abu Dhabi at the time of the survey (72.7%), followed by Dubai-based respondents (16.4%). Participants with a good educational background were selected to obtain a greater warranty of results. Out of all the participants, 45.0% had a bachelor's degree, followed by 26.1% with a master's degree. Ph.D. degree holders contributed 8% of the total participants. As the prison facilities are

owned and operated by the government in UAE, most of the participants were from the public sector (92.0%) with only 6.3% respondents from the private sector. 33.6% hold a higher position in their respective departments (MIES), followed by 28.2% from PCI. 68.9% of participants had more than 10 years of relevant experience.

**Table 1.** Sociodemographic characteristics of the respondents participating in the survey.

| Characteristics | Categories | Frequency | Percent |
|---|---|---|---|
| Gender | Male | 202 | 84.9 |
| | Female | 36 | 15.1 |
| Age | ≤30 years | 49 | 20.6 |
| | 31–40 years | 120 | 50.4 |
| | 41–50 years | 54 | 22.7 |
| | 51–60 years | 13 | 5.5 |
| | ≥60 years | 2 | 0.8 |
| Nationality | UAE | 207 | 87.0 |
| | Foreigners | 31 | 13.0 |
| Emirates | Abu Dhabi | 173 | 72.7 |
| | Dubai | 39 | 16.4 |
| | Sharjah | 11 | 4.6 |
| | Ajman | 2 | 0.8 |
| | Fujairah | 1 | 0.4 |
| | Umm Al Quwain | 3 | 1.3 |
| | Ras Al Khaimah | 9 | 3.8 |
| Qualification | College diploma | 10 | 4.2 |
| | Bachelor's degree | 107 | 45.0 |
| | Master's degree | 62 | 26.1 |
| | PhD | 19 | 8.0 |
| | Others | 40 | 16.8 |
| Organization | Government | 219 | 92.0 |
| | Private | 15 | 6.3 |
| | Other | 4 | 1.7 |
| Affiliation | Ministry of Interior | 45 | 18.9 |
| | Punitive and Correctional Institutions | 67 | 28.2 |
| | Ministry of Interior Engineers and Specialists department | 80 | 33.6 |
| | Prisons Consultants, Experts, and Architects | 36 | 15.1 |
| | Strategic Partners | 10 | 4.2 |
| Experience | ≤3 years | 11 | 4.6 |
| | 4–6 years | 29 | 12.2 |
| | 7–9 years | 34 | 14.3 |
| | ≥10 years | 164 | 68.9 |

*3.2. Normality and Reliability Analysis*

The data for key selection criteria and smart building technologies being used in UAE were subjected to a normality test. The Shapiro-Wilk test of normality showed that the data for key selection criteria and smart building technologies were not normally distributed ($p < 0.05$) (Table 2). The reliability analysis was performed to verify the authenticity of the responses provided by the participants. Cronbach's alpha for the tested characteristics understudy was higher than 0.80 confirming the participant's deep knowledge about the questions asked. The Cronbach's alpha for various selection criteria of smart building technologies was 0.910, and for smart building technologies currently used in the UAE prisons, a value of 0.913 was recorded (Table 2).

**Table 2.** Shapiro-Wilk test of normality and reliability analysis.

| Variable | W | Skewness | Kurtosis | DF | *p*-Value | Cronbach's Alpha |
|---|---|---|---|---|---|---|
| Selection criteria | 0.895 | −1.509 ± 0.158 | 5.907 ± 0.314 | 238 | 0.000 | 0.910 |
| Smart building technologies | 0.923 | −1.019 ± 0.158 | 3.842 ± 0.314 | 238 | 0.000 | 0.913 |

Where W = Shapiro-Wilk test statistic.

### 3.3. Smart Building Technologies Currently Being Used in UAE Prisons

Of the 15 smart building technologies currently used in the UAE prisons, the result in Table 3 shows that all these smart building technologies have mean scores above 4.0 (very important), except the E-shopping system. The top five smart building technologies include fire protection system, video surveillance system, wiring infrastructure, HVAC system and audio-video system. As for the opinions of different groups (public vs. private respondents) are concerned, the Mann-Whitney U test shows non-significant differences for 8 out of 15 smart building technologies by comparing government and private sector respondents. In general, the public respondents perceived higher importance to most of the smart building technologies than private respondents, except for building management system (BMS). Government respondents when compared to private respondents for "Fire Protection System" demonstrated no statistical difference ($p = 0.2768$) with a negligible effect size ($r = 0.0711$), for "Video Surveillance System" has no statistical difference ($p = 0.0778$) with a negligible-small effect size ($r = 0.1153$), for "Wiring Infrastructure" no statistical difference ($p = 0.4912$) with a negligible effect size ($r = 0.0450$), and for "HVAC" no statistical difference ($p = 0.1735$) with a negligible effect size ($r = 0.0890$), while for "Networking System" showed statistical difference ($p = 0.0026$) with a small effect size ($r = 0.1968$) followed by "Advanced Information System" ($p = 0.0062$) with negligible-small effect size ($r = 0.1788$), "Building Management System" ($p = 0.0058$) with small effect size ($r = 0.1804$) and "Energy and Sustainability System" ($p = 0.0002$) with small effect size ($r = 0.2462$) (Table 3).

### 3.4. Key Selection Criteria

Regarding the 14 selection criteria of smart building technology for prison buildings, the results revealed that all the listed selection criteria were rated as "very important" with a mean value of more than 4.0 (Table 4). The top five selection criteria were safety and security of the smart building technology (4.55), the anti-hacking capability of the smart building technology (4.46), the high working efficiency of the smart building technology (4.37), easy access to spare parts (4.32) and strict compliance with international standards (4.32), followed by allowing for further upgrade (4.29) as, without the ability to upgrade, the smart building technology will become void soon with the advancement in technology. The respondents were also concerned that the smart building technology should be durable (4.27) with a longer life with affordable maintenance support (4.22), and it must be compatible (4.25) with the existing prison buildings and new prison design. Environmental sustainability (4.25) has been an issue for all the prison buildings, so was ranked as very important as was cost-effectiveness (4.27). In UAE, very few companies perform business with smart building technologies, so the government has to rely on international firms for the purchase of new smart building technology. For suppliers operating in UAE, it was highlighted that their firms should have good credibility (4.11) to meet the standards and should have a wide range of products with authentic brand and warranty (4.09) (Table 4).

**Table 3.** Descriptive statistics, Mann-Whitney U test, and Cohan's effect size for the smart building technologies being used in the UAE prisons.

| Smart Building Technologies | Mean | Rank | Govt (*n* = 219) | | Private (*n* = 15) | | Mann Whitney U | Z | *p*-Value | *r* |
|---|---|---|---|---|---|---|---|---|---|---|
| | | | Mean | Rank | Mean | Rank | | | | |
| Fire protection system | 4.66 | 1 | 4.66 | 1 | 4.60 | 1 | 1389.500 | −1.0876 | 0.2768 | 0.0711 |
| Video surveillance system | 4.63 | 2 | 4.64 | 2 | 4.40 | 3 | 1228.000 | −1.7638 | 0.0778 | 0.1153 |
| Wiring infrastructure | 4.45 | 3 | 4.48 | 3 | 4.40 | 3 | 1482.500 | −0.6885 | 0.4912 | 0.0450 |
| Heat, ventilation, and air-conditioning system (HVAC) | 4.39 | 4 | 4.40 | 4 | 4.20 | 4 | 1332.500 | −1.3609 | 0.1735 | 0.0890 |
| Audio-video system | 4.33 | 5 | 4.36 | 5 | 3.80 | 6 | 1461.000 | −0.9155 | 0.3599 | 0.0598 |
| Networking system | 4.31 | 6 | 4.33 | 6 | 3.80 | 6 | 922.500 | −3.0108 | 0.0026 | 0.1968 |
| Building automation system (BAS) | 4.27 | 7 | 4.28 | 8 | 4.07 | 5 | 1250.500 | −1.9365 | 0.0528 | 0.1266 |
| Advanced information system | 4.27 | 7 | 4.30 | 7 | 3.73 | 7 | 998.000 | −2.7351 | 0.0062 | 0.1788 |
| Building management system (BMS) | 4.27 | 8 | 4.24 | 9 | 4.47 | 2 | 982.500 | −2.7595 | 0.0058 | 0.1804 |
| Lighting system | 4.20 | 9 | 4.22 | 11 | 3.80 | 6 | 1225.500 | −1.7939 | 0.0728 | 0.1173 |
| Energy and sustainability system | 4.20 | 9 | 4.24 | 10 | 3.53 | 8 | 744.000 | −3.7660 | 0.0002 | 0.2462 |
| E-messaging system | 4.07 | 10 | 4.12 | 12 | 3.07 | 10 | 1050.000 | −2.5597 | 0.0105 | 0.1673 |
| Vertical transportation system | 4.06 | 11 | 4.10 | 13 | 3.33 | 9 | 1054.500 | −2.5532 | 0.0107 | 0.1669 |
| Funds transfer system | 4.03 | 12 | 4.07 | 14 | 3.33 | 9 | 894.500 | −3.0584 | 0.0022 | 0.1999 |
| E-shopping system | 3.48 | 13 | 3.55 | 15 | 2.47 | 11 | 1623.000 | −0.0881 | 0.9298 | 0.0058 |

Where *r* = Cohan's effect size.

**Table 4.** Descriptive statistics, Mann-Whitney U test, and Cohan's effect size of the key selection criteria for selecting smart building technologies for UAE prisons.

| Selection Criteria | Mean | Rank | Govt (*n* = 219) | | Private (*n* = 15) | | Mann-Whitney U | Z | *p*-Value | *r* |
|---|---|---|---|---|---|---|---|---|---|---|
| | | | Mean | Rank | Mean | Rank | | | | |
| Safety and security of the smart building technology | 4.55 | 1 | 4.56 | 1 | 4.40 | 3 | 1641.000 | −0.0067 | 0.9946 | 0.0004 |
| The anti-hacking capability of new smart building technology | 4.46 | 2 | 4.45 | 2 | 4.53 | 1 | 1562.500 | −0.3476 | 0.7282 | 0.0227 |
| High working efficiency of new smart building technology | 4.37 | 3 | 4.37 | 3 | 4.27 | 4 | 1377.500 | −1.1464 | 0.2516 | 0.0749 |
| Easy access (availability) to spare parts | 4.32 | 4 | 4.30 | 5 | 4.53 | 1 | 1616.000 | −0.1162 | 0.9075 | 0.0076 |
| Strict compliance with international standards | 4.32 | 5 | 4.31 | 4 | 4.40 | 3 | 1515.000 | −0.6018 | 0.5473 | 0.0393 |
| Allow for further upgrade | 4.29 | 6 | 4.28 | 7 | 4.47 | 2 | 1293.000 | −1.4956 | 0.1348 | 0.0978 |
| The durability of new smart building technology | 4.27 | 7 | 4.29 | 6 | 3.87 | 6 | 1359.500 | −1.2141 | 0.2247 | 0.0794 |
| Suitability of the new smart building technology to the existing buildings | 4.25 | 8 | 4.27 | 8 | 3.87 | 6 | 1273.500 | −1.5837 | 0.1133 | 0.1035 |
| Compatibility of new smart building technology with the design of new prisons | 4.25 | 8 | 4.26 | 9 | 4.07 | 5 | 1492.500 | −0.6460 | 0.5183 | 0.0422 |
| Affordable maintenance support | 4.22 | 9 | 4.20 | 11 | 4.40 | 3 | 1325.500 | −1.3571 | 0.1748 | 0.0887 |
| Environmental sustainability of the new smart building technology | 4.21 | 10 | 4.24 | 10 | 3.80 | 7 | 1248.500 | −1.6611 | 0.0967 | 0.1086 |
| Cost-effectiveness | 4.17 | 11 | 4.18 | 12 | 3.87 | 6 | 1073.500 | −2.4290 | 0.0151 | 0.1588 |
| Supplier's credibility in the market | 4.11 | 12 | 4.11 | 13 | 4.07 | 5 | 1582.500 | −0.2533 | 0.8000 | 0.0166 |
| Smart building technology brand and warranty | 4.10 | 13 | 4.07 | 14 | 4.47 | 2 | 1499.000 | −0.6216 | 0.5342 | 0.0406 |

Where *r* = Cohan's effect size.

Similarly, respondents working in government and private organizations have contrasting opinions about the importance of selection criteria. The Mann-Whitney U test revealed non-significant statistical differences ($p > 0.05$) for all the tested selection criteria when comparing government and private sector respondents, while cost-effectiveness proved to be the only selection criteria that demonstrated statistically significant difference ($p = 0.0151$) with negligible effect size ($r = 0.1588$). The government respondents emphasized higher ratings for safety and security with no statistical difference ($p = 0.9946$) and negligible effect size ($r = 0.0004$), high working efficiency with non-significant difference (0.2516) and negligible effect size ($r = 0.0749$), durability of new smart building technology having non-significant difference ($p = 0.2247$) with negligible effect size ($r = 0.0794$), compatibility with the design of new prisons showing no statistical difference ($p = 0.5183$) with negligible effect size ($r = 0.0422$), sustainability demonstrated no statistical difference ($p = 0.1133$) with negligible effect size ($r = 0.1035$), and suppliers' credibility illustrated no statistical difference ($p = 0.8000$) with negligible effect size ($r = 0.0166$) whereas other criteria were ranked higher by the private sector respondents. The anti-hacking capability was the top-ranked selection criterion for the private group (statistically non-significant difference having $p = 0.7282$ with negligible effect size r = 0.0227) followed by allowing for further upgrades (statistically non-significant difference having $p = 0.1348$ with negligible effect size r = 0.0978) and smart building brand and warranty (statistically non-significant difference having $p = 0.5342$ with negligible effect size r = 0.0406). The public respondents ranked all criteria above the mean value of 4.00 except for durability, suability, sustainability, and cost-effectiveness. (Table 4).

*3.5. Exploratory Factor Analysis*

The factor analysis extracted three factors based on the Eigenvalues following varimax rotation. The first factor denoted as economic criteria included five selection criteria that cumulatively contributed 18.063% variation to overall variance (Table 5). The selection criteria in this factor included durability of the new smart building technology with a factor loading of 0.695 followed by the suitability of the new smart building technology to the existing buildings (0.665), compatibility of new smart building technology with the design of new prisons (0.532), affordable maintenance support (0.528), and installation and operational costs (0.471). The second factor assigned as performance-related criteria contributed 16.532% variation and has the highest factor loading of 0.659 for the high working efficiency of new smart building technology. The other criteria in this factor included strict compliance to international standards (0.620), the anti-hacking capability of new smart building technology (0.613), easy access (availability) to spare parts (0.521), and safety and security of the smart building technology (0.467). The third factor sustainability-related criteria demonstrated a 15.991% variance and included four key criteria such as smart building technology brand and warranty with a factor loading of 0.720, followed by suppliers' credibility in the market (0.626), allowing for further upgrade (0.612) and environmental sustainability of the new smart building technology (0.412). The scree plot for all the extracted factors based on their Eigenvalues has been illustrated in Figure 1. The scree plot provides a quick indication of the elbow-shaped curve of the extracted factors based on Eigenvalues. Only three factors had Eigenvalues greater than 1, whereas the rest of the factors were valued below the marginal level of 1.

**Table 5.** Explanatory factor analysis for the key selection criteria of smart building technologies for UAE prisons.

| Factor 1 Economic Criteria | Factor Loading | Covariance (%) |
|---|---|---|
| The durability of new smart building technology | 0.695 | 18.063 |
| Suitability of the new smart building technology to the existing buildings | 0.665 | |
| Compatibility of new smart building technology with the design of new prisons | 0.532 | |
| Affordable maintenance support | 0.528 | |
| Installation and operational costs | 0.471 | |

**Table 5.** *Cont.*

| Factor 2 Performance-related criteria | | |
|---|---|---|
| High working efficiency of new smart building technology | 0.659 | 16.532 |
| Strict compliance with international standards | 0.620 | |
| The anti-hacking capability of new smart building technology | 0.613 | |
| Easy access (availability) to spare parts | 0.521 | |
| Safety and security of the smart building technology | 0.467 | |
| **Factor 3 sustainability-related criteria** | | |
| Smart building technology brand and warranty | 0.720 | 15.991 |
| Supplier's credibility in the market | 0.626 | |
| Allow for further upgrade | 0.612 | |
| Environmental sustainability of the new smart building technology | 0.420 | |

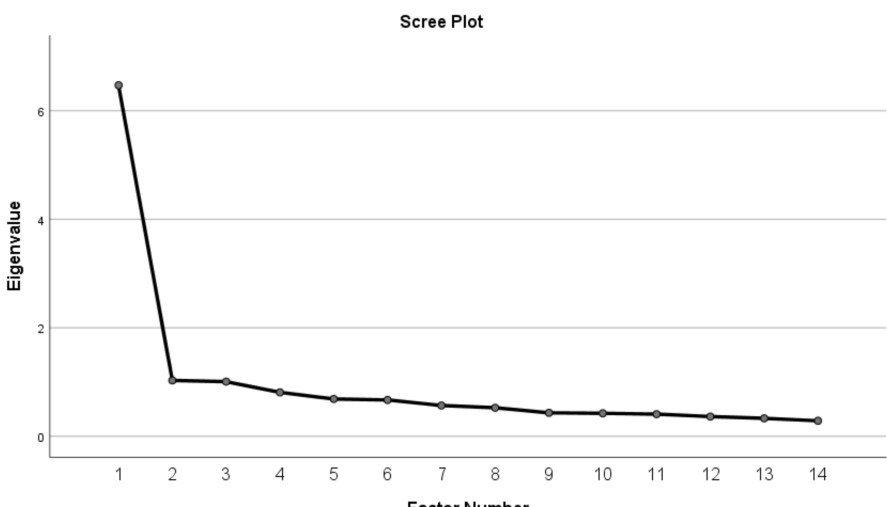

**Figure 1.** Scree plot of the key selection criteria for the selection of smart building technologies in the UAE prisons.

## 4. Discussion

### 4.1. Factor 1: Economic Criteria

The first factor is comprised of economic selection criteria that accounts for 18.063% variation for the tested selection criteria. This factor possessed prominent criteria, including the durability of new smart building technology, the suitability of the new smart building technology to the existing buildings, compatibility of new smart building technology with the design of new prisons, affordable maintenance support, and installation and operational costs.

Many previous studies have emphasized the high cost of smart building technologies. These costs are not only limited to purchase cost but also accounts for installation costs, operational costs, and maintenance costs. Operational and maintenance costs are key selection criteria that may be considered while selecting smart building technologies. High management costs are the key hurdles in the way of adopting smart building technologies as reported by [32]. Very high procurement costs of smart building technology hinder its adoption at a large scale, as the building owners do not have enough resources to purchase smart technologies [33]. In comparison with small buildings, operational costs are usually lower for larger buildings that have advanced BAS. Contrastingly, redundant technologies such as smart thermostats are more economical for use in smart buildings. This comparison of buildings with advanced BAS does not seem true when compared with wireless cloud-based EIS. Tracy [34] reported that cloud-based EIS requires 30% fewer installation costs than a typical BAS. Similarly, cloud computing technologies provide not only economic comfort to the users but are also safer and more reliable than traditional data management

systems [35]. Panchalingam and Chan [36] reported that artificial intelligence (AI)-powered smart building technologies are cost-effective, have higher security benefits and energy savings, and enhance the life span of the devices compared to traditional smart technologies. Many studies have also focussed on the issue of lowering maintenance costs of smart building technologies, thus enhancing the longer life span of smart technology [37,38].

### 4.2. Factor 2: Performance-Related Criteria

The next factor, performance-related criteria, explained for 16.532% of the variation including criteria such as high working efficiency, strict compliance to international standards, the anti-hacking capability of new smart building technology, easy access to spare parts, and safety and security of the smart building technologies.

The performance of smart building technology is mainly assessed by speed, accuracy, and working efficiency. Performance is not only attributed to the smartness of the building system but also a measure of the level of satisfaction of end-users [39]. The Performance indicators may include high working efficiency, ability to integrate with BMS, automated monitoring, energy saving, reliability, and compatibility with other building systems [40]. Cohen, et al. [41] highlighted five indicators to assess whether either smart building fulfills the needs of its residents. The prominent indicators include user comfort, residents' physical well-being, an appropriate lightning system, noise control system, and quality air. According to Zhang and Hu [32], the lack of modern security and management technologies may lead to poor performance of the building systems whereas managing the maintenance costs, effective building management through digital infrastructure tools, and adopting appropriate smart building technologies boost the overall performance of the smart building system [7].

Evidence exists in the literature on adopting smart building technologies for enhancing the energy performance of smart buildings. For instance, the Siemens Middle East Headquarters in Abu Dhabi has managed to reduce its power consumption by 63% by adopting smart building concepts [42]. In Amsterdam, The Edge building has been able to achieve a 70% reduction in electricity consumption by adopting a smart energy management system that adjusts temperature and light levels according to the building operations and occupancy [43]. Theoretically, Moreno, et al. [44] have shown that smart buildings can achieve up to 98.5% energy expectation maximization (EM) efficiency in variable occupancy. Another study by Ożadowicz [45] investigated the impact of building automation on energy efficiency by analyzing a university building. The study revealed that electric energy use was reduced by 36.6%, while heat energy declined by 30.5% due to the implementation of the automation project.

### 4.3. Factor 3: Sustainability-Related Criteria

The remaining selection criteria were grouped under sustainability-related criteria for smart building technology which contributed 15.991% to the total variance for all the tested selection criteria. The critical selection criteria placed in this group include the brand and warranty of the smart building technology, suppliers' credibility in the market, ability of the smart buildings to allow for further upgrades, and environmental sustainability of the smart building technology.

Environmental sustainability has been a key criterion for smart building systems to ensure less energy consumption, less waste generation, and low greenhouse gas emissions. Smart buildings are aimed firsthand to keep the environment sustainable but constructing sustainable buildings is not an easy task. Serious efforts were implemented in the recent past to make the environment sustainable by improving working efficiency and eliminating waste through modern building systems. Smart buildings could be environmentally sustainable by reducing greenhouse emissions, enhancing recycling, and adopting renewable energy sources [42,43,46]. For this purpose, in 2014, the Dubai Municipality issued a directive requiring the use of Building Information Modeling (BIM). All buildings of over 300,000 square feet or 40 stories were required to use the BIM [47]. The Masdar smart

city, a project worth $22 billion, was designed with sustainable urban technology ideas to ensure zero carbon and sustainable settlement [48]. Most cities and towns have adopted sustainability as a long-term solution to growth and development. Thus, governments particularly in the developed world have promoted the concept of sustainable design [49]. For instance, in the United States, the number of energy and environmentally efficient buildings has increased due to the tax benefits associated with such buildings [50]. In recent times, sustainability has been associated with green buildings whereas performance is related to smart buildings because green buildings serve the primary purpose of limiting environmental impacts whereas smart buildings focus on the efficient operation of the building system [51].

AlWaer and Clements–Croome [52] have outlined nine key performance indicators (KPIs) to obtain sustainability in terms of the energy efficiency of smart buildings. The critical KPIs included maximizing passive solar energy, enhancing electrical energy conservation, and conserving non-renewable energy resources. A smart building's energy efficiency and carbon reduction ability can be assessed by adopting six KPIs outlined by Global Sustainability Perspective from Jones Lang LaSalle (JLL). These KPIs include wireless sensors, cloud computing, advanced communication system, data analytics tool, remote control system and integrated administrative system [53].

## 5. Conclusions

This paper identifies and evaluates the key selection criteria for smart building technologies used in UAE prisons. The findings demonstrated that all the tested selection criteria had mean values above 4 (very important). The highly ranked selection criteria include safety and security, antihacking capability, the high working efficiency of the smart building technology, easy access to spare parts, and strict compliance with international standards, followed by allowing for further upgrades. Concerning the importance of smart building technologies currently used in the UAE prisons, the prominent technologies include fire protection system, vertical transportation system, wiring infrastructure, HVAC, and lighting systems, each with a mean ranking of more than 4.0 (very important), except E-shopping system. The exploratory factor analysis (EFA) reveals that three potential factors were extracted that contributed approximately 50% variation. These factors were named as economic, performance-related, and sustainability-related criteria, each factor contributing approximately 17% variation.

This study generated key insights about potential smart building technologies being implemented in prison buildings in the UAE. First, various critical selection criteria were highlighted in this study that could serve as the basis of guidelines for the broader audience ranging from academia to private stakeholders. Secondly, the potential smart building technologies were listed that could be used for prison facilities helping prison administration with ease to control inmates and operate prison matters with very less manpower engagement.

This study suffers the following limitations. First, the selection criteria identified are mainly applicable to the prison structures of the UAE only. Similar studies can be conducted in other geographical locations in gulf countries where prison structures are similar to that of the UAE. Another limitation of this study is that the findings of the current survey might be more useful for public organizations, because most of the respondents are from the public sector. Private organizations do not avail much liberty in sharing ideas in government policy matters regarding the implementation of smart building technologies.

**Author Contributions:** B.X. designed the experiment and reviewed the manuscript; M.A.M.M.A. conducted the survey, performed data analysis and wrot e the manuscript; M.N. helped in the revision of the manuscript and data analysis. All authors have read and agreed to the published version of the manuscript.

**Funding:** This research was funded by Abu Dhabi Police GHQ on behalf of the development of punitive and correctional facilities in the UAE. The funders had no role in the study design, data collection, and analysis, decision to publish, or preparation of the manuscript.

**Institutional Review Board Statement:** The study was conducted in accordance with the Declaration of Helsinki, and approved by the Ethics Committee of QUEENSLAND UNIVERSITY OF TECHNOL-OGY, BRISBANE, AUSTRALIA (protocol code 2000000835 and date of approval 22/01/2021).

**Informed Consent Statement:** Informed consent was obtained from all subjects involved in the study.

**Data Availability Statement:** The data protection rights are reserved by QUT.

**Acknowledgments:** The authors are highly indebted to Abu Dhabi Police GHQ for providing the funds for this study. Also, special thanks to Muhammad Farooq, QUT for his help in the data analysis and editing of this manuscript. The author is thankful to the supervisory committee of QUT for their continuous support throughout this research study.

**Conflicts of Interest:** The authors declare no conflict of interest.

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
