# Peer review of "Identifying Key Selection Criteria for Smart Building Technologies in the United Arab Emirates Prisons"

_buildings, doi:10.3390/buildings12081171_

Round 1

Reviewer 1 Report

1. The affiliation of the second author should be added.

2. I cannot see the necessity of including "2.2. English and Arabic Translations".

3. Lines 175-176, KMO instead of Kaiser-Meyer-Olkin (KMO) should be used, same on line 252. Could you clarify the meaning of “values of 0.920 and 0.898”?

4. Figure 2 should be further explained.

5. How would the study be helpful for a broad audience? A major challenge is that there are strong architectural constraints due to the fact that the design of prison buildings is primarily governed by security concerns. The measures usually developed to meet thermal comfort and IAQ etc. requirements are not always applicable to the prison environment.

Round 2

Reviewer 2 Report

The manuscript has been greatly improved. However, there is some misunderstanding on the part of the reviewer.

In general

There are at least to conceptual questions: (1) parametric versus nonparametric statistics when Likert data should be analyzed and (2) the relationship between statistically independent units and statistically dependent units?

(1)   Likert data are ordinal data. Typically, assumption of normality for Likert data is not met. Shapiro–Wilk test results for all data should be reported. The authors use parametric statistics (for example, Principal Component Analysis (PCA). PCA includes the use of Pearson’s correlation. However, before using the Pearson’s correlation, four Gauss-Markov assumptions need to be tested. Authors must provide a rigorous proof of the application of parametric statistics. Authors must also provide rigorous evidence why nonparametric statistics cannot be applied in this study.

(2)   Historically, when describing statistical packages, the term "observations" has been used to describe both statistically independent units and statistically dependent units. For example, the paired t-test or the Wilcoxon signed-rank test can be used to assess the statistical effect in the same group before and after treatment. This group contains ten individuals (i.e., statistically independent units). However, each individual was measured twice, “before” and “after” i.e., there are twenty statistically dependent subunits. 

In specific

Line 168. For the nonparametric effect size, N is number of submits, not number of participants statistically independent units.

Line 169. What is N equal to?

Tables 2 and 3 contain an unequal number of persons in "Govt" (n = 219) and "private" (n = 15). In this context, statistical “paired models” cannot be used. To compere between "Govt" (n = 219) and "private" (n = 15), the Wilcoxon–Mann–Whitney test and Cliff’s δ effect size should be used.

An interpretation of the effect size in terms of "small/medium/large" should be presented in the "Method" section.

Round 3

Reviewer 2 Report

In general

Written interaction between the authors of the manuscript and the reviewer showed that the output statistics contain a huge number of errors. However, if this manuscript contains only descriptive statistics, then this study will be of value.

The results of the Shapiro-Wilk test are not presented in the manuscript.

An interpretation of the effect size is not provided in the manuscript.

The subsection of effect size interpretation should contain the following information: The r value varies from 0 to close to 1. The interpretation values for r commonly are: 0.10 ≤ 0.3 (small effect), 0.30 ≤ 0.5 (moderate effect) and ≥ 0.5 (large effect).

Lines 14-15. “Wilcoxon signed-rank test was conducted”. However.

Line 169. It was written “Mann-Whitney U test”.

Line 174 “n= number of cases”. What are “cases”?

Line 177. It was written “Exploratory Factor Analysis (EFA)”. However.

Line 190. It was written “non-parametric factor extraction method.

However, EFA is one of non-parametric factor extraction method”

Line 185. Bartlett's Test of sphericity is an indicator to identify equal variances [44].

Bartlett's Test of sphericity is not correct procedure for testing equality of variances.

For this purpose, the Bartlett’s Test for Equality of Variance should be used.

The following is a summary of the Bartlett's Test of sphericity.

Bartlett’s Test of Sphericity compares an observed correlation matrix to the identity matrix. Essentially it checks to see if there is a certain redundancy between the variables that we can summarize with a few numbers of factors.

The null hypothesis of the test is that the variables are orthogonal, i.e. not correlated. The alternative hypothesis is that the variables are not orthogonal, i.e. they are correlated enough to where the correlation matrix diverges significantly from the identity matrix.

This test is often performed before we use a data reduction technique such as principal component analysis or factor analysis to verify that a data reduction technique can actually compress the data in a meaningful way.

Note: Bartlett’s Test of Sphericity is not the same as Bartlett’s Test for Equality of Variances. This is a common confusion, since the two have similar names. 

Lines 188-189. The procedure "Generalized Least Squares" refers to parametric methods. The reference [46] used non-parametric multivariate analysis to evaluated latent associations between different variables.

Lines 225-228 and Table 2. It is written “the Mann-Whitney U test shows significant differences with small effect sizes by comparing government and private sector respondents for all the tested smart building technologies except fire protection system, wiring infrastructure, HVAC, audio-video system, lighting system and E-shopping system.”

In Table 2, the two groups were compared using Z, p-value, and effect size.

The two groups did not statistically differ from each other in the following parameters: “Fire protection system”, “Wiring infrastructure”, ”HVAC”, “Audio-video system”, and “E-shopping system”.

The content of the Table 2 is fundamentally different from the text.

Why is Z negative?

Exact p values to four decimal places should be presented in the Table 2.

The effect size is misinterpreted.

“r value of the effect size” should be presented in the Table 2.

In addition, Mann-Whitney U values must be presented in the Table 2. This is necessary for the transparency of the statistical procedure.

Lines 234-265 and table 3. Statistical analysis of Table 3 is missing from the manuscript.

Why is Z negative?

Exact p values to four decimal places should be presented in the Table 3.

The effect size is misinterpreted.

“r value of the effect size” should be presented in the Table 3.

In addition, Mann-Whitney U values must be presented in the Table 3. This is necessary for the transparency of the statistical procedure.

Lines 177-179. It was written “Exploratory Factor Analysis (EFA) was performed on the data obtained from each variable to extract the latent factors which contributed significant contributions to variation among the tested variables”.

However, neither the results nor the conclusions considered or discussed "latent factors".

Round 4

Reviewer 2 Report

In general

Statistical terminology has been moderately improved. However, the interpretation of Table 3 and 4 contains statistical errors.

In Specific

Line 174. Why was the lower bound on the small effect size (r = 0.1) ignored? Give an exact quotation from Cohen's paper [42]. 

Lines 229-234 and Table 3. It was written “…the Mann-Whitney U test shows significant differences by comparing government [ reviewer group 1] and private [reviewer, group 2] sector respondents for all the tested smart building technologies except fire protection system, video surveillance system, wiring infrastructure, HVAC, audio-video system, building automation system, lighting system and E-shopping system. The effect size coefficient demonstrated small effect sizes (r≤0.3) for all the tested smart building technologies”.

However, Table 3 provides the following statistics: group 1 compared to group 2 for "Fire Protection System" has no statistical difference (p = 0.2768) with a negligible effect size (r = 0.0711), for “Video surveillance system” has no statistical difference (p = 0.0778) with a negligible-small effect size (r = 0.1153), for “Wiring infrastructure” has no statistical difference (p = 0.4912) with a negligible effect size (r = 0.0450), for “HVAC” has no statistical difference (p = 0.1735) with a negligible effect size (r = 0.0890), while for “Networking system” has statistical difference (p = 0.0026) with a small effect size (r = 0.1968) and so on. Therefore, 8 of the 15 variables have no statistical difference and these differences have a negligible effect size.

Tables 3 and 4 contain an analysis of descriptive and inferential statistics. However, the text of the manuscript does not contain an analysis of the inference statistics (p-value and effect size).

The titles of tables 3 and 4 should contain more complete information. 

It should be noted that the p-value can tell the reader if an effect exists, while the effect size shows a significant value.

Author Response

Thank you very much for your patient guidance.

Please see the attached responses.

Round 5

Reviewer 2 Report

The manuscript can be accepted.